# Modulating the CXCR2 Signaling Axis Using Engineered Chemokine Fusion Proteins to Disrupt Myeloid Cell Infiltration in Pancreatic Cancer

**DOI:** 10.3390/biom15050645

**Published:** 2025-04-30

**Authors:** Benjamin N. Christopher, Lena Golick, Ashton Basar, Leticia Reyes, Reeder M. Robinson, Aaron O. Angerstein, Carsten Krieg, G. Aaron Hobbs, Denis C. Guttridge, John P. O’Bryan, Nathan G. Dolloff

**Affiliations:** 1Department of Pharmacology and Immunology, Medical University of South Carolina, Charleston, SC 29425, USA; christob@musc.edu (B.N.C.); golick@musc.edu (L.G.); basar@musc.edu (A.B.); reyesl@musc.edu (L.R.); robinree@musc.edu (R.M.R.); angerste@musc.edu (A.O.A.); 2Department of Pathology and Laboratory Medicine, Medical University of South Carolina, Charleston, SC 29425, USA; kriegc@musc.edu; 3Department of Biochemistry, Medical University of South Carolina, Charleston, SC 29425, USA; hobbsg@musc.edu (G.A.H.); obryanjo@musc.edu (J.P.O.); 4MUSC Hollings Cancer Center, Charleston, SC 29425, USA; guttridg@musc.edu; 5MUSC Darby Children’s Research Institute, Charleston, SC 29425, USA; 6Department of Pediatrics, Medical University of South Carolina, Charleston, SC 29425, USA; 7Zucker Institute for Innovation Commercialization, Charleston, SC 29425, USA

**Keywords:** pancreatic ductal adenocarcinoma, tumor microenvironment, immunosuppression, myeloid cell migration, MDSCs, CXCR2, CXCL1

## Abstract

Pancreatic ductal adenocarcinoma (PDAC) has one of the lowest 5-year survival rates of all cancers, and limited treatment options exist. Immunotherapy is effective in some cancer types, but the immunosuppressive tumor microenvironment (TME) of PDAC is a barrier to effective immunotherapy. CXCR2+ myeloid-derived suppressor cells (MDSCs) are abundant in PDAC tumors in humans and in mouse models. MDSCs suppress effector cell function, making them attractive targets for restoring anti-tumor immunity. In this study, we show that the most abundant soluble factors released from a genetically diverse set of human and mouse PDAC cells are CXCR2 ligands, including CXCL8, CXCL5, and CXCL1. Expression of CXCR2 ligands is at least partially dependent on mutant KRAS and NFκB signaling, which are two of the most commonly dysregulated pathways in PDAC. We show that MDSCs are the most prevalent immune cells in PDAC tumors. MDSCs expressed high levels of CXCR2, and we found that myeloid cells readily migrate toward conditioned media (CM) prepared from PDAC cultures. We designed CXCR2 ligand-Fc fusion proteins to modulate the CXCR2 chemotactic signaling axis. Unexpectedly, these fusion proteins were superior to native chemokines in binding and activation of CXCR2 on myeloid cells. These “superkines” were potent inhibitors of PDAC CM-induced myeloid cell migration and were superior to CXCR2 small-molecule inhibitors and neutralizing antibodies. Our findings suggest that CXCR2 superkines may disrupt myeloid cell recruitment to PDAC tumors, ultimately improving immunotherapy outcomes in patients with PDAC.

## 1. Introduction

Pancreatic ductal adenocarcinoma (PDAC) is a highly lethal disease with a 5-year survival rate of less than 10% [1]. PDAC exhibits significant resistance to therapy, mediated largely by a dense desmoplastic stroma and an immunosuppressive tumor microenvironment (TME) [2]. Immune checkpoint inhibitors (ICIs) have had success in a variety of cancer types, but their efficacy in PDAC has been disappointing. Moreover, ICIs are only approved for the 1–2% of patients that have high microsatellite instability (MSI-H) or DNA mismatch repair deficiency (dMMR) [3]. The PDAC TME supports an abundance of immunosuppressive immune and stromal cells such as myeloid-derived suppressor cells (MDSCs) [4], tumor-associated macrophages (TAMs) [5], regulatory T cells (Tregs) [6], and cancer-associated fibroblasts (CAFs) [7]. Consequently, infiltration and activation of anti-tumor immune cells such as CD8+ T cells and natural killer (NK) cells is limited [8,9,10]. The formation and maintenance of the PDAC TME are primarily driven by KRAS mutation, which is present in approximately 90% of PDAC cases. KRAS signaling in PDAC enhances the secretion of cytokines and chemokines, which recruit immunosuppressive cells and foster a supportive environment for immune evasion and tumor progression [11].

Myeloid cells play a distinctive role in shaping the PDAC TME due to their inflammatory properties, plasticity, and role in immunosuppression. In healthy individuals, the myeloid compartment plays a vital role in the inflammatory response, making it essential for proper immune system function and wound healing. However, the cytokines produced by PDAC have a profound effect on the myeloid compartment, which promotes a chronic inflammatory state where the inflammatory response never reaches a resolution phase [12,13]. This dysregulation underlies the characterization of tumors as “wounds that do not heal” [14].

One of the most significant consequences of this dysregulation is the expansion of MDSCs, which are potent suppressors of the anti-tumor immune response [15,16]. MDSCs employ multiple mechanisms to inhibit CD8+ T cells, such as depleting arginine, an essential amino acid for proper T cell proliferation and activation [17,18]; secreting reactive oxygen species, inducing stress in T cells and inactivating the T cell receptor [19,20]; and downregulating L-selectin on T cells, hindering T cell trafficking to lymphoid organs, which is essential for the T cell response [21]. Additionally, MDSCs suppress the activity of NK cells [22], promote the expansion of immunosuppressive regulatory T cells [23,24], and promote the immunosuppressive polarization of macrophages [25]. Beyond promoting the expansion of MDSCs, PDAC attracts MDSCs to the TME through chemokine secretion. Chemokine receptors play a central role in immune cell migration by directing cells to tissues that secrete their corresponding ligands. This migration relies on the ability of cells to sense a chemokine gradient and coordinate movement towards areas of higher chemokine concentration [26,27].

CXC motif chemokine receptor 2 (CXCR2) is one of many chemokine receptors implicated in establishing the immunosuppressive PDAC TME. In healthy individuals, the CXCR2 axis is critical for the homing of neutrophils and other cell types to sites of injury. This process is coordinated by the secretion of CXCR2 ligands from macrophages and mast cells in proximity to the site of injury [28]. However, PDAC and many other cancers hijack the normal physiological roles of CXCR2 ligands to recruit MDSCs and protect themselves from the anti-tumor immune response. The expression of CXCR2 on MDSCs is critical for their recruitment to the PDAC TME [29]. Since MDSCs are significant contributors to the immunosuppressive PDAC TME and CXCR2 is important for their recruitment to the PDAC TME, CXCR2 is an attractive target for pancreatic cancer. In our study we developed an IgG1 Fc-based fusion protein to inhibit myeloid cell migration towards PDAC-derived factors by hyperactivating the CXCR2 axis.

IgG Fc-based fusion proteins are a widely utilized class of biologics designed to enhance the stability, half-life, and efficacy of protein-based therapeutics. These molecules consist of a protein, usually a receptor or ligand, fused to the Fc domain of immunoglobulin G (IgG), which extends serum half-life by recycling through the neonatal Fc receptor (FcRn) and improves structural stability [30]. The Fc domain may retain its native function, engaging Fc receptors on immune cells to modulate immune responses, or it may be engineered to eliminate Fc receptor interactions [31]. Several clinically relevant IgG Fc fusion proteins, including etanercept (TNFR-Fc) and aflibercept (VEGFR-Fc), have been developed for inflammatory diseases and oncology, where they modulate immune activation, block immunosuppressive pathways, or reshape the tumor microenvironment [32,33]. These therapeutics work by acting as cytokine traps to sequester pro-inflammatory or pro-angiogenic ligands, thereby preventing their interaction with cell surface receptors and dampening pathological signaling within the inflammatory or tumor microenvironment [34]. Other Fc-based fusion proteins incorporate ligands as their fusion partners and act as activators or suppressors of immune responses [31]. In this study, we identified an unanticipated consequence of Fc fusion, where the addition of the Fc domain enhanced the binding and signaling activity of CXCL1, indicating that Fc fusion can actively influence cytokine function rather than serving solely as a pharmacokinetic enhancer, thereby revealing new opportunities for therapeutic development.

## 2. Materials and Methods

### 2.1. Cell Lines and Reagents

Cell lines were purchased from the American Tissue Culture Collection (Manassas, VA, USA) and cultured according to the manufacturer’s specifications. The cell lines were PANC-1, CFPAC-1, BxPC-3, Capan-1, Capan-2, HEK293, THP-1, HL60, and K562. HPNE cells with doxycycline-inducible KRAS mutations were a gift from Dr. Aaron Hobbs (Medical University of South Carolina). Cells were grown at 37 °C and in 5% CO_2_ in the recommended base media supplemented with heat-inactivated fetal bovine serum (FBS) (900–108, GeminiBio, Sacramento, CA, USA), and the antibiotics 1% penicillin (10,000 units/mL) and 1% streptomycin (10,000 µg/mL) were purchased from Selleck Chemicals (Houston, TX, USA). Other materials were as follows: SX-682 (S8947), Navarixin (S8506), and AZD5069 (S6645) were purchased from Selleck Chemicals (Houston, TX, USA), αCXCR2 (MA1-24669, Thermo Fisher Scientific, Waltham, MA, USA), doxycycline (D9891, Sigma-Aldrich, St. Louis, MO, USA), IκB-SR retrovirus was a gift from Dr. Dennis Guttridge (Medical University of South Carolina), and TNF-α (1006-050, CellGenix, Freiburg, Germany). A Proteome Profiler Human XL Cytokine Array Kit (ARY022B, R & D Systems, Minneapolis, MN, USA) was used according to the manufacturer’s specifications. All ELISAs were performed according to the manufacturer’s specifications. Kits were as follows: Human CXCL1 (88-52122-22, Invitrogen, Waltham, MA, USA), Human CXCL5 (dy254-05, R & D Systems, Minneapolis, MN, USA), Human CXCL8 (88-8086-22, Invitrogen, Waltham, MA, USA), Mouse CXCL1 (P352782, R & D Systems, Minneapolis, MN, USA), and Mouse CXCL5 (MX000, R & D Systems, Minneapolis, MN, USA).

### 2.2. Subcloning

Genes for CXCL1 (35–107), CXCL5 (37–114), IgG1 Fc (216–447 Eu numbering), and CXCR2 were codon-optimized for human cells by GenScript Biotech (Piscataway, NJ, USA). CXCL1 or CXCL5 were fused to Fc with DNA encoding GGGGS. CXCR2 was cloned into pLJM1 using Afe1 and EcoR1. All other genes were subcloned into a custom pcDNA3.4 using NheI and XhoI with an N-terminal secretion signal and a C-terminal 6x-His tag. Proper ligation and fusion of each gene was confirmed by DNA sequencing at Eurofins Genomics (Louisville, KY, USA).

### 2.3. Protein Expression and Purification

CXCL1, CXCL5, Fc, and all fused genes were expressed in the expi293F expression system (Thermo Fisher Scientific, Waltham, MA, USA) following the manufacturer’s protocol. Briefly, 1 µg/mL pcDNA3.4 containing each gene with a secretion signal and C-terminal 6x-His tag was transfected into expi293F cells using expifectamine 293. Medium was collected by centrifugation of cells 5–7 days later when cell viability was <60%. Medium was then loaded at 5 mL/min onto a 5 mL HisTrap Excel column (Cytiva, Marlborough, MA, USA) equilibrated with buffer A (20 mM sodium phosphate, 500 mM NaCl, pH 7.4). After loading, the column was washed 3 subsequent times with buffer A containing 10 mM, 25 mM, and then 50 mM imidazole (A10221, Thermo Fisher Scientific, Waltham, MA, USA). The bound protein was subsequently eluted with buffer A containing 500 mM imidazole. Eluted proteins were then dialyzed overnight in PBS, at pH 7.4 at 4 °C, sterile-filtered, flash-frozen in liquid nitrogen, and stored at −80 °C. Protein purity was determined using SDS-PAGE. Concentrations were determined using a Bradford assay using Bio-Rad Protein Assay Dye Reagent Concentrate (Bio-Rad Laboratories, Hercules, CA, USA) diluted 1:5 in MilliQ water.

### 2.4. Western Blotting

Cells were collected on ice, rinsed with cold 1× PBS, centrifuged at 2500 rpm at 4 °C for 5 min, and lysed in 1x cell lysis buffer (Cell Signaling Technology, Danvers, MA, USA) with the addition of protease (Thermo Fisher Scientific, Waltham, MA, USA) and phosphatase (Thermo Fisher Scientific, Waltham, MA, USA) inhibitors. For Western blots analyzing the nuclear fraction, cytoplasmic and nuclear fractions were isolated with an NE-PER Nuclear and Cytoplasmic Extraction Kit (78833, Thermo Fisher Scientific, Waltham, MA, USA). Cell lysates were clarified, and their relative protein concentrations were determined via a Bradford assay. Values were normalized to the lowest average absorbance at 595 nm. Gel samples were prepared by mixing lysates with SDS sample buffer containing β-mercaptoethanol (bp176-100, Thermo Fisher Scientific, Waltham, MA, USA) (final concentration 1×) or without β-mercaptoethanol for non-reducing conditions and 1x cell lysis buffer. Samples were boiled for 10 min, loaded on NuPAGE Bis-Tris Gel 4–12% (Invitrogen, Waltham, MA, USA), and subjected to gel electrophoresis at 55 mA for 1 h and 30 min in 1× NuPAGE MOPS SDS Running Buffer (Invitrogen, Waltham, MA, USA). Gels were transferred to polyvinylidene difluoride (PDVF) membranes at 300 mA for 2 h in 1x transfer buffer containing 25 mM trizma base, 192 mM glycine, and 20% methanol. PVDF membranes were blocked for 1 h at room temperature with 5% milk in TBS-Tween prior to incubation with primary antibodies in 5% milk in TBS-Tween overnight at 4 °C. Primary antibodies were as follows: KRAS was a gift from Dr. Aaron Hobbs (Medical University of South Carolina) (WH0003845MI, Sigma-Aldrich, St. Louis, MO, USA), β-Actin (A5441, Sigma-Aldrich, St. Louis, MO, USA), IκBα (9242, Cell Signaling Technology, Danvers, MA, USA), α-Tubulin (3873, Cell Signaling Technology, Danvers, MA, USA), NF-κB p65 (sc-8008, Santa Cruz Biotechnology, Dallas, TX, USA) Phospho-NF-κB p65 (3033, Cell Signaling Technology, Danvers, MA, USA), Lamin A/C (2032 Cell Signaling Technology, Danvers, MA, USA), ERK1/2 (4695 Cell Signaling Technology, Danvers, MA, USA), Phospho-ERK1/2 (4377 Cell Signaling Technology, Danvers, MA, USA), AKT (pan) (4685 Cell Signaling Technology, Danvers, MA, USA), and Phospho-AKT (4060 Cell Signaling Technology, Danvers, MA, USA). Membranes were washed with 1× TBS-Tween and subsequently incubated with secondary antibodies in 5% milk in TBS-Tween for 2 h at room temperature. Secondary antibodies conjugated to horseradish peroxidase were goat anti-mouse IgG-H+L (31430, Invitrogen, Waltham, MA, USA) and goat anti-rabbit IgG-H+L. Detection was finalized using ECL (T32209, Thermo Fisher Scientific, Waltham, MA, USA) or Super Signal (34094, Thermo Fisher Scientific, Waltham, MA, USA) detection reagents.

### 2.5. Migration Assays

CM for migration assays was obtained by plating 10 × 10^6^ CFPAC-1 or PANC-1 cells in 10 mL of fully supplemented medium in 10 cm^2^ petri dishes, incubating for 48 h, and collecting the medium. THP-1 cells were collected and pelleted at 300× *g* for 5 min. Cells were resuspended in fully supplemented RPMI-1640, and 5 × 10^5^ cells were added in 0.5 mL with or without treatment to ThinCert Cell Culture Inserts (24 well, 8 μm pore size) (662638, Greiner Bio-One, Kremsmünster, Austria). Next, 0.75 mL of PDAC CM or fully supplemented DMEM was added the lower chamber in each well. Plates were allowed to incubate for 2 h at 37 °C with 5% CO_2_. After incubation, the inserts were removed, and 0.25 mL of CellTiter-Glo (Promega, Madison, WI, USA) was added to each well. Plates were shaken and incubated at 37 °C for 10 min. After incubation, 0.2 mL of the mixture in each well was transferred to black 96-well plates in technical duplicate. Luminescence was recorded on a SpectraMax L Microplate Reader (Molecular Devices, San Jose, CA, USA) at 470 nm with a 1 s integration time.

### 2.6. Thermodynamic Stability Assay

The thermodynamic stability of Fc, CXCL1-Fc, and each CXCL1-Fc variant was determined using a Tycho NanoTemper (Munich, Germany). Proteins were diluted to 100 µg/mL in PBS and drawn into glass capillary tubes. The protein in the glass capillary tubes was then subjected to a thermal ramp, and the absorbance at 350 nm and 330 nm was monitored. The first derivative of the ratio of 350 nm/330 nm was then plotted as a function of temperature by the onboard software, where the peak was defined as the melting temperature (T_m_) of the sample.

### 2.7. Aggregation Assay

Proteins were diluted to 86 µg/mL and analyzed using a PROTEOSTAT Protein aggregation assay following the manufacturer’s instructions (Enzo, Cat #ENZ-51023-KP050). The plate was read using an excitation setting of 550 nm and an emission filter of 600 nm.

### 2.8. Production of Lentiviral Particles and Infection of K562 Cells

A total of 2.5 × 10^6^ HEK-293T cells were plated in 10 mL of DMEM with 10% FBS. Cells were infected with 15 μg pLJM1 CXCR2 DNA, 7.5 μg pCMV-dR8.91, and 1.5 μg pCMV-VSV-G diluted in Opti-MEM and Lipofectamine 2000. Seventy-two hours post-transfection, the supernatant containing viral particles was collected and stored at −80 °C. A total of 5 × 10^5^ K562 cells were infected with 2 mL of supernatant and 1 mL of RPMI with 10% FBS with 8 μg/mL polybrene. Cells were treated with 2 μg/mL puromycin (A1113803, Thermo Fisher Scientific, Waltham, MA, USA) for selection of infected cells. Experiments were performed once stable expression was confirmed by flow cytometry.

### 2.9. Flow Cytometry

Cells were collected and washed with ice-cold FACS buffer (PBS with 1% FBS and 150 μM CaCl_2_). Fc receptors were blocked using Human SeroBlock (BUF070, Bio-Rad Laboratories, Hercules, CA, USA) or Mouse SeroBlock (BUF041 Bio-Rad Laboratories, Hercules, CA, USA) according to the manufacturer’s instructions. Cells were then treated with fluorophore-conjugated antibodies at dilutions indicated by the manufacturers and incubated at 4 °C for 30 min. Cells were then washed with 1 mL of FACS buffer and resuspended in 300 µL of FACS buffer and analyzed with either the FITC or PE channel or both of a NovoCyte flow cytometer (Acea Biosciences; San Diego, CA, USA). Antibodies used for flow cytometry were CD11b-FITC (24442, Cell Signaling Technology, Danvers, MA, USA), Rat IgG2b, κ-FITC (400634, BioLegend, San Diego, CA, USA), GR-1-PE (78355S, Cell Signaling Technology, Danvers, MA, USA), Rat IgG2b-PE (27426, Cell Signaling Technology, Danvers, MA, USA) mouse CXCR2-FITC (149310, BioLegend, San Diego, CA, USA), Rat IgG2a, κ-FITC (553929, BD Pharmingen, Franklin Lakes, NJ, USA), His-Tag-Alexa Fluor 488 (14930 Cell Signaling Technology, Danvers, MA, USA), human CXCR2-PE (12-1892-42, Thermo Fisher Scientific, Waltham, MA, USA), and Mouse IgG1, κ-PE (400114, BioLegend, San Diego, CA, USA).

### 2.10. MDSC Isolation from Mouse Spleens

C57BL/6 mice were injected subcutaneously with 100,000 KPC908 cells in 0.1 mL of a 1:1 mixture of DMEM and Matrigel (354234, Corning, Corning, NY, USA) or a mixture with no cells as a control. Tumors were implanted in the left and right flanks of each mouse for a total of two tumors per mouse. Spleens were harvested from the C57BL/6 mice once tumors were near 2 cm^3^. Spleens were disrupted in PBS containing 2% FBS by the rubber head of a syringe plunger and passed through a 70 μm mesh nylon strainer (542070 Greiner Bio-One, Kremsmünster, Austria). Cells were then centrifuged at 300× *g* for 10 min and resuspended in PBS containing 2% FBS and 1 mM EDTA (J15694.AE Thermo Fisher Scientific, Waltham, MA, USA). Mouse MDSCs were then isolated by negative selection from the single-cell suspension using an EasySep™ Mouse MDSC (CD11b+Gr1+) Isolation Kit (STEMCELL Technologies; Vancouver, BC, Canada). Proper isolation was verified by flow cytometry. Mice were handled according to the Medical University of South Carolina (MUSC) institutional animal care and use committee (IACUC).

### 2.11. CyTOF

To comprehensively characterize immune cell populations within the KPC tumor microenvironment, mass cytometry by time-of-flight (CyTOF) was performed on barcoded CD45+ immune cells isolated from subcutaneous tumors. Single-cell suspensions were prepared and stained with a 20+ metal-tagged antibody panel designed to distinguish key immune subsets, including myeloid-derived suppressor cells (MDSCs), tumor-associated macrophages (TAMs), neutrophils, and T cells. Cells were then fixed, permeabilized for nuclear staining, and acquired on a Helios™ mass cytometer (Standard Biotools, South San Francisco, CA, USA). Data were normalized using bead-based calibration and analyzed using the CATALYST workflow [35,36]. Dimensionality reduction techniques, including UMAP, were applied to visualize immune cell clustering and heterogeneity.

## 3. Results

### 3.1. CXCR2 Ligands Are the Most Abundant Factors Secreted by PDAC Cells

The PDAC tumor microenvironment is composed of multiple immune cell types and a dense network of stromal fibroblasts and proteins [9,37,38]. The formation of this desmoplastic stroma is coordinated by a host of signaling molecules including cytokines, growth factors, and chemokines that are released from tumor cells, stromal fibroblasts, and infiltrating immune cells [13,39]. To identify the secreted factors produced solely by PDAC tumor cells, we screened the secretome of a genetically diverse panel of human PDAC cells. We measured the relative concentration of 105 soluble factors in the conditioned medium of four cell lines (PANC-1, BxPC3, Capan1, and CFPAC) using an antibody-based multiplex screening strategy. We found the highest levels of CXCL8/IL-8, CXCL5/ENA-78, GDF-15, VEGF, and CXCL1/GROα to be present in the conditioned media (CM) of all four cell lines examined (Figure 1A,B and Appendix A). Interestingly, three of the five secreted factors that were detected in all four cell lines were CXCR2 ligands (CXCL1, CXCL5, and CXCL8). In addition to these five, we observed a high expression of several other potentially relevant molecules in CM from two out of four or three out of four but not all of the cell lines tested. These included angiogenin, lipocalin-2, serpinE1, Dkk1, uPAR, and IL-17a (Figure 1B). Given the high levels and consistent expression of CXCR2 ligands in PDAC cells, we focused on these chemokines and confirmed the initial screening findings by quantitative ELISA (Figure 1C). Lower levels of all three were detected in immortalized normal pancreatic epithelial cells (HPNE) and HEK cells, demonstrating that the release of CXCR2 ligands is related to the specific genotype and phenotype of PDAC cells rather than a consequence of tissue culture conditions or methods (e.g., serum, media supplements, etc.). Elevated levels of CXCL8 were detected in Capan-1, CFPAC-1, and PANC-1; CXCL5 in Capan-1 and CFPAC-1; and CXCL1 in Capan-1, CFPAC-1, and PANC-1. Mouse-specific CXCR2 ligands, CXCL5 and CXCL1 (mice lack the CXCL8 gene or a human homolog), were also abundant in the CM from murine PDAC cell lines including Panc02 cells and two cell lines derived from PDAC tumors in Kras+/LSL-G12D;Trp53+/LSL-R172H;Pdx-1-Cre (KPC) mice (Figure 1D), further supporting the observation that CXCR2 ligand secretion is a characteristic of the PDAC phenotype.

### 3.2. CXCR2 Ligand Production in PDAC Cells Is Induced by Mutant KRAS and NFκB Signaling

KRAS mutations are one of the most common genomic abnormalities in PDAC and are known to be driver events [40]. The cell autonomous effects of KRAS mutations have been extensively studied, but their impact effects on the secretome are less understood. Therefore, we next conducted experiments to determine if KRAS mutations also contribute to heightened CXCR2 ligand secretion by PDAC. We first used a model of immortalized normal human pancreatic epithelial (HPNE) cells, which were engineered with a doxycycline (DOX)-inducible system to express KRAS G12D and G12R mutants (Figure 2A). CM from HPNE cultures revealed that the KRAS mutations led to significant increases in CXCL8 and CXCL5 secretion but not to changes in CXCL1 production (Figure 2B). We note that the absolute magnitude of CXCL5 expression induced by KRAS mutations was 10-fold lower than the levels detected in human PDAC cell lines like Capan-1 (75–100 pg/mL versus ~1000 pg/mL). In human PDAC cell lines, which harbor KRAS mutations as well as a host of other genomic alterations, we found that treatment with the KRAS G12D mutant specific inhibitor, MRTX1133, significantly suppressed CXCR2 ligand secretion in G12D mutant cells (PANC-1 [41]; Figure 2C). In support of the KRAS G12D specificity of MRTX1133, the drug had no effect on KRAS G12V mutant cells (CFPAC-1 [42]; Figure 2C). These findings partially implicate mutant KRAS signaling in the induction and secretion of CXCR2 ligands, with the caveat that KRAS mutations contribute to, but are not sufficient to, induce all CXCR2 ligands.

We also investigated the role of nuclear factor kappa light chain enhancer of activated B cells (NF-κB) in CXCR2 ligand production. NF-κB is a linchpin regulator of pro-inflammatory cytokine and chemokine production [43,44], and dysregulation of NF-κB signaling has been reported in approximately 70% of PDAC cases [45,46]. We explored the role of NF-κB by overexpressing an IκB super repressor (IκB-SR), which is a degradation-resistant form of IκBα. Activation of NFκB is set in motion by cytokines such as TNFα that induce IκBα degradation and the release and nuclear translocation of NFκB transcription factor subunits. IκB-SR is stable following TNFα treatment and prevents NFκB nuclear translocation by binding and sequestering it in the cytoplasm (Figure 2D). In PANC-1 cells, we found that IκB-SR significantly reduced CXCL1 levels in CM (Figure 2E) but had no effect on CXCL8 or CXCL5 production and secretion. Similar to what we observed with KRAS mutations, these findings suggest a multifactorial regulation of CXCR2 chemokines by a variety of factors and parallel signaling pathways. Our results show that these include mutant KRAS and NFκB signaling, which are some of the most commonly deregulated pathways in PDAC.

### 3.3. CXCR2 Is Part of the Immunophenotype of Myeloid-Derived Suppressor Cells (MDSCs) in Mice with PDAC

We next analyzed the cellular components of the PDAC tumor immune microenvironment in KPC mice. To achieve this, we conducted CyTOF on CD45-positive immune cells isolated from subcutaneous tumors established from a KPC clone. The vast majority (>80%, N = 4) of infiltrating immune cells were myeloid in origin (Figure 3A,B), as indicated by T cell-negative (CD4, CD8), CD45+, Ly6C+, and CD11b+ cells with varying levels of F4/80, Ly6G, CD11c, and MHC-II (Appendix A). Three distinct myeloid subsets were defined based on differential marker expression: Myeloid 1 cells exhibited low CD11b, Ly6G, CD11c, and f4/80 with intermediate Ly6C and high MHC-II, consistent with inflammatory monocytes or immature monocytic cells; Myeloid 2 cells expressed low levels of Ly6C and MHC-II with intermediate Ly6G, CD11c, CD11b, and F4/80, indicative of tumor-associated macrophages (TAMs) or differentiated macrophages; and Myeloid 3 cells demonstrated high CD11c, MHC-II, CD11b, Ly6C, Ly6G, and F4/80 consistent with antigen-presenting dendritic-like cells. The most abundant of those myeloid cell clusters (Myeloid 3) corresponded to an immunophenotype consistent with MDSCs [CD11b- and GR1 (Ly6C/G)-positive] [47].

We confirmed the presence of tumor-infiltrating MDSCs in PDAC tumors growing in mice by analyzing CD11b+GR1+ cells by flow cytometry (Figure 3C). We also analyzed the spleens from tumor-naïve and tumor-bearing mice, and in line with previous reports by others [47], we found that spleens from PDAC-tumor-bearing mice contained a significantly higher portion of MDSCs (Figure 3D). Importantly, splenic MDSCs also expressed CXCR2 (Figure 3E). Altogether, these findings implicate CXCR2 and its ligands in the high density of MDSCs found in the PDAC TME.

### 3.4. Design and Characterization of CXCR2 Ligand Fusion Proteins

We next set out to engineer CXCR2-targeted biotherapeutic molecules that could modulate this pathway and disrupt myeloid cell trafficking to PDAC tumor sites. We designed, expressed, and purified CXCL5 and CXCL1 fusion proteins that consisted of the mature chemokines conjugated to the N-terminal hinge domain of an IgG1 fragment crystallizable (Fc) region. A flexible GGGGS linker was used, and the Fc domain was engineered with multiple site-specific mutations. These included (1) C220S, which eliminates crosslinking of Fc domains by removing the cysteine that is normally involved in an interchain disulfide bond with the light chain of IgG1, (2) N297G, which eliminates antibody-dependent cellular cytotoxicity (ADCC) by removing a glycosylation site that is required for FcγR binding [48], and (3) the combination of M428L and N434S, which increases serum half-life in vivo through enhanced affinity for neonatal Fc receptors (FcRn) [49]. We gave priority to CXCL5 and CXCL1 over CXCL8, since the former are selective CXCR2 ligands, whereas the latter binds with similar affinity for both CXCR1 and CXCR2 and therefore offered less specificity. The layouts of the proteins produced for this study are shown in the insert of Figure 4A. Analysis of the protein products by SDS-PAGE revealed that the Fc fusion proteins formed dimers, presumably due to Fc domain coupling (Figure 4A). The CXCL5-Fc and CXCL1-Fc fusion proteins showed greater binding than the native chemokines to THP-1 cells (Figure 4B), which are acute myeloid leukemia (AML) cells of myeloid origin and CXCR2-positive (Appendix A). Similar results were observed with HL60 AML cells and primary human neutrophils (Figure 4C), which were CXCR2-positive (Appendix A). Binding was CXCR2-dependent, as CXCL5-Fc bound to CXCR2-positive K562 human chronic myelogenous leukemia (CML) cells but not to isogenic CXCR2-negative K562 cells (Appendix A). Ultimately, CXCL1-Fc was chosen for further analysis due to its greater binding to THP-1 cells over a range of concentrations (Figure 4D). In line with superior cell surface binding of the chemokine-fusion proteins compared to the monomeric non-fusions, the Fc fusions also induced the most intense cell signaling in THP-1 cells. For example, CXCL1 induced weak and transient induction of the p42/44 MAPK (ERK) and Akt pathways, whereas the CXCL1-Fc super chemokine (dubbed “superkines” by others [50]) stimulated these pathways with greater intensity and for sustained time periods in both THP-1 and HL60 cells (Figure 4E). Similar results were observed in dose range experiments, where CXCL1-Fc generated stronger ERK signaling than the native CXCL1 across a range of concentrations (Figure 4F).

### 3.5. CXCR2 Ligand Fusion Protein Disrupts Chemotaxis of Myeloid Cells

Previous studies have shown that knockout of the mouse homolog of CXCR2 (Cxcr2) reduces infiltration of MDSCs, neutrophils, and tumor-associated macrophages into mouse PDAC tumors [51,52]. CXCR1/2 small-molecule inhibitors were reported to reduce myeloid cell infiltration into tumors in mice [53,54]. However, others observed compensatory effects that counteract this response [55], which is perhaps not unexpected given the multifactorial composition of PDAC-secreted factors and the receptor/ligand promiscuity common among chemokines [56]. Given these challenges, we took an alternative approach to disrupting CXCR2 signaling. Rather than inhibiting CXCR2, we hypothesized that we could disrupt chemotactic gradients by hyperactivating the CXCR2 signaling axis with our ligand fusion superkine. Because the binding and signaling through these molecules was superior to the native chemokines, we reasoned that these signals would cancel out directional chemotactic signals. We first demonstrated that CM from multiple PDAC cell lines indeed induced migration of myeloid cells (THP-1) in a Boyden chamber transwell assay (Figure 5A). When CXCL1-F_c_ was added to the top well of the chamber with cells, it significantly reduced their ability to migrate towards the PDAC CM chemotactic gradient (Figure 5B). Interestingly, native CXCL1 had no effect on migration, further demonstrating the inferiority of the natural chemokine compared to the engineered fusion protein. Dose–response analyses were also conducted to quantify these differences. The migration Inhibitory Concentration 50 (IC50) for CXCL1-Fc was calculated at 8.5 nM compared to >1000 nM for CXCL1 and the Fc control protein (Figure 5C). In addition, we found that an anti-CXCR2 neutralizing antibody, like the native chemokine, failed to inhibit THP-1 myeloid cell migration toward PDAC CM. Further to this point, we found that multiple small-molecule CXCR2 inhibitors (SX-682, AZD5069, and Navarixin) also failed to inhibit THP-1 cell migration at concentrations that inhibited CXCL8- and CXCL5-induced signaling (Figure 5D and Appendix A). Together, these findings support our hypothesis that hyperactivation of the CXCR2 signaling axis with a CXCL1 fusion superkine is able to overcome chemotactic forces in PDAC CM, whereas CXCR2 inhibitors are not. We also found that inhibition by CXCL1-Fc could be neutralized by a CXCR2 antibody (Figure 5E). This demonstrates that there are CXCR2-independent chemotactic forces in PDAC CM, and while CXCR2 inhibition is not sufficient to inhibit migration, the effects of CXCL1-Fc remain dependent on binding and signaling through CXCR2.

### 3.6. Structural Investigation of CXCL1-Fc

We initially constructed CXCR2 ligand Fc fusions to improve the stability of the chemokines and ultimately improve the pharmacokinetic properties, as reported by others [57]. However, having observed stark differences in binding, signaling, and cellular effects between the native and fusion proteins, we next asked why these differences existed. We considered two hypotheses. One was that the forced dimerization of the chemokine by virtue of Fc conjugation could enhance signaling in an unexpected non-linear fashion, or, secondly, the Fc domain of the fusion could impart enhanced signaling independent of the chemokine dimer. To test this, we made two structural variants of CXCL1-Fc by introducing knobs-into-holes technology, where complementary mutations (T366Y and Y407T) were made in the CH3 domain of the Fc region to prevent Fc homodimerization, as is typically carried out for bispecific antibody construction [58,59]. The first (CXCL1^1^-Fc^2^) consisted of a CXCL1-Fc monomer with a knob mutation matched to an Fc monomer with a hole mutation, forming an intact Fc dimer with only one CXCL1 protein per molecule (Figure 6A). The second (CXCL1^1^-Fc^1^) consisted of a CXCL1-Fc monomer with a knob mutation but no complementary hole mutation counterpart, resulting in a dimerization-deficient CXCL1-Fc with monomeric chemokine and Fc region (Figure 6A). With these constructs in hand, we were able to test the importance of CXCL1 bivalency (CXCL1^1^-Fc^2^) and Fc bivalency (CXCL1^1^-Fc^1^) in the superior activity of the CXCL1-Fc superkine. In the binding assays, CXCL1-Fc bound significantly more effectively than either of the variants with an approximately three- to four-fold higher mean fluorescence intensity per cell, with no significant difference in binding between the two monomeric CXCL1 variants. However, both variants bound better than CXCL1 alone (Figure 6B,C). Signaling experiments revealed that CXCL1-Fc was the most potent molecule of the three, as it induced sustained ERK signaling compared to the two variants, which were transient, peaking at 1–2 min and rapidly returning to baseline. Again, both variants were superior to CXCL1 alone (Figure 6D). However, we noted that the CXCL1-Fc knob/hole variant achieved a significantly higher signal amplitude than the CXCL1-Fc knob that was comparable to the full CXCL1-Fc protein. This suggested that the intact Fc dimer had an impact on the downstream signaling activity of the molecule even though it did not necessarily increase its binding properties. Finally, the migration experiments revealed that CXCL1-Fc was a superior inhibitor of myeloid cell migration toward a PDAC CM gradient (Figure 6E), as it had a significantly lower IC50 compared to CXCL1^1^-Fc^2^ (8.5 nM vs. 28 nM). CXCL1^1^-Fc^1^ was the least effective inhibitor of myeloid cell migration, with an IC50 25-fold and 7.5-fold higher than CXCL1-Fc (8.5 nM vs. 211 nM) and CXCL1^1^-Fc^2^ (28 nM vs. 211 nM, Figure 6E). This demonstrates that fusion of the CXCL1 chemokine to an Fc domain, and to an Fc dimer in particular, is important for CXCL1-Fc-mediated inhibition of myeloid cell migration. To better understand the contribution of the Fc domain to the potency of the CXCL1-Fc superkine, we investigated the biochemical properties of the different protein variants. In the thermal stability studies, we found that the melting temperature (T_m_) of CXCL1^1^-Fc^1^ was ~2 °C lower than CXCL1-Fc and CXCL1^1^-Fc^2^ (Appendix A). Additionally, assessments of protein aggregation revealed that the CXCL1^1^-Fc^1^ variant was more prone to aggregation than the other proteins (Appendix A). These results suggest that the inferiority of CXCL1^1^-Fc^1^ may be due to decreased stability from the loss of Fc dimerization. Taken together, these results demonstrate that fusion of CXCL1 to an Fc domain is sufficient to enhance CXCL1 binding and signaling, independent of Fc or CXCL1 dimerization, thereby enabling CXCL1-Fc to function as an inhibitor of myeloid cell migration.

## 4. Discussion

MDSCs are a major constituent of the PDAC TME and represent a significant obstacle to anti-tumor immunity and immunotherapies due to their suppressive effects on immune effector cell types. In our study, we identified CXCR2 ligands as the most abundant soluble factors secreted from PDAC cells and that CD11b+/GR1+ MDSCs express the receptor for these ligands, CXCR2. Additionally, we showed that mutant KRAS and NFκB signaling, which are two of the major aberrant signaling pathways in PDAC, contributed to the production and secretion of CXCR2 ligands from human and mouse PDAC cells. In addition to the production of these ligands from PDAC cells, previous studies have reported the secretion of CXCR2 ligands from cancer-associated fibroblasts (CAFs) [39,60] and tumor-associated macrophages (TAMs) [61] in the PDAC TME. Our data indicate that myeloid cells, and MDSCs specifically, were the most prominent immune cell type in KPC tumors. These MDSCs expressed CXCR2 and were present in the tumors and enriched in the spleens of these mice. Taken together, our data and previously published data strongly demonstrate that CXCR2 ligands are produced in high quantity from PDAC tumors and promote the recruitment and/or retention and differentiation of CXCR2+ MDSCs, which counteract anti-tumor immunity.

Several CXCR1/2 small-molecule inhibitors have shown efficacy in preclinical models of inflammatory diseases and in combination treatments for cancer [62,63]. However, their results in clinical trials have been mixed with few positive signals of efficacy [64,65]. The development of therapeutics that inhibit MDSCs or myeloid cell recruitment to PDAC tumors presents several challenges. Myeloid cell migration is driven by multiple chemokines and chemokine receptors. While CXCR2 ligands are the most prevalent soluble factors secreted by a genetically diverse set of PDAC cells, CXCR2 is not the only chemokine receptor that regulates myeloid cell migration. For example, work by others has shown that CC Chemokine Receptor 2 (CCR2) also mediates MDSC recruitment in PDAC [66]. Chemokine networks also show a high degree of binding promiscuity between ligands and receptors [56]. CXCL8, for example, which is one of the most prevalent PDAC ligands, binds to both CXCR2 and CXCR1, and CXCR2 is also a receptor for CXCL1, -2, -3, -5, and -7 [67]. Therefore, pharmacological inhibition of a single chemokine receptor like CXCR2 using a small molecule or neutralizing antibody is likely to be offset by compensatory recruitment of other chemokine receptor signals.

To address this challenge, we hypothesized that hyperactivating CXCR2, rather than inhibiting it, could circumvent the complexity and compensatory nature of chemokine networks by disrupting directional signaling created by chemokine gradients. This was made possible by the serendipitous discovery of the CXCL1-Fc superkine that demonstrated enhanced CXCR2 signaling properties. Chemokines induce cell migration by concentrating chemokine receptor signaling at the cellular pole facing a chemokine gradient. This enriched signaling biases cytoskeletal dynamics at the leading edge of the cell and propels the cell in the direction of the chemokine source [68]. We propose that the hyperactivation of CXCR2 by the CXCL1-Fc superkine neutralizes chemotactic signals originating from PDAC factors and inhibits migration by preventing cytoskeletal polarization and directional movement. Because of this CXCR2 hyperactivation effect, CXCL1-Fc could impair not just movement toward CXCR2 ligand gradients but also broadly inhibit chemotaxis toward chemokine gradients in general. In support of this hypothesis, we showed that CXCL1-Fc is the only molecule that could inhibit myeloid cell migration towards PDAC CM. CXCR1/2 small-molecule inhibitors were unable to block migration, as was a CXCR2-neutralizing antibody. The native CXCL1 chemokine was also unable to inhibit migration, demonstrating the importance of the Fc domain in the design of the fusion superkine.

Beyond PDAC, CXCL1-Fc may have broader therapeutic applications. Given its superior ability to inhibit myeloid cell migration compared to small-molecule inhibitors, it may be beneficial in treating inflammatory diseases. In addition, CXCL1-Fc may have applications in wound healing. The CXCR2 axis is critical for neutrophil recruitment and epithelial resurfacing mediated by keratinocytes in the wound healing process, and loss of CXCR2 expression delays wound healing [69]. Various strategies have been explored to enhance wound healing by modulating chemokine signaling. Notably, a topical therapy utilizing lactic acid bacteria engineered to express CXCL12 was recently found to accelerate wound healing in a phase 1 clinical trial [70]. Given that CXCL1-Fc exhibits superior CXCR2 activation compared to CXCL1 alone, it may represent an alternative or complementary strategy for enhancing neutrophil recruitment and keratinocyte migration during wound healing. Furthermore, other chemokines may benefit from Fc fusion conjugation, broadening the potential for therapeutic intervention across various conditions.

IgG Fc fusion proteins are widely used in biomedical applications to enhance stability, solubility, and functional activity in diverse biomedical applications [71]. In our study, we leveraged the IgG Fc region with the intent to improve pharmacokinetic properties, and in the process, we unexpectedly created a CXCL1 superkine with enhanced binding to CXCR2 and enhanced activation of CXCR2 downstream signaling effectors like ERK (p42/44 MAPK) and Akt. Our data show that this is not due simply to the forced dimerization of CXCL1, as a CXCL1-dimerization-deficient variant still demonstrated superior function. This finding suggests that the Fc region itself contributes to the enhanced activity of the fusion. Potential explanations for this enhanced function include Fc-mediated alteration of CXCL1 steric positioning, stabilization of a CXCL1 conformation that promotes CXCR2 activation, or enabling CXCL1 to behave as a partial agonist, selectively activating specific components of the CXCR2 signaling pathway while sparing others. Further studies are needed to elucidate the precise mechanism and to explore the translational potential of CXCL1-Fc in preclinical studies of PDAC and other disease models.

## 5. Conclusions

Collectively, our data implicate the CXCR2 signaling axis in the migration of myeloid cells toward soluble factors released by PDAC tumors (Figure 7A). CXCR2+ myeloid cells infiltrate PDAC tumors along a chemotactic gradient that is composed predominantly of CXCR2 ligands, including CXCL1, CXCL5, and CXCL8. These myeloid cells appear to have suppressor function (i.e., MDSCs) prior to entering the PDAC environment, as our data show that PDAC tumors increase MDSC numbers in secondary lymphoid organs such as the spleen. Once in the PDAC tumor, MDSCs suppress immune effector cell function and shield PDAC cells from host immune surveillance. Through design and evaluation of a CXCL1-Fc fusion “superkine” with unexpectedly potent signaling activity, we found that hyperstimulation of CXCR2 of MDSCs could completely block migration toward PDAC-secreted factors (Figure 7B). This approach was, somewhat paradoxically, more effective than inhibiting CXCR2 with small-molecule inhibitors or anti-CXCR2 antibodies. This suggests that there are additional chemotactic factors in PDAC CM that can compensate for CXCR2 inhibition and attract myeloid cells in the presence of the inhibitors. The CXCL1-Fc superkine, on the other hand, had the ability to drown out all other chemotactic gradients due to its superior signaling ability, and it completely inhibited myeloid cell migration.

## Figures and Tables

**Figure 1 biomolecules-15-00645-f001:**
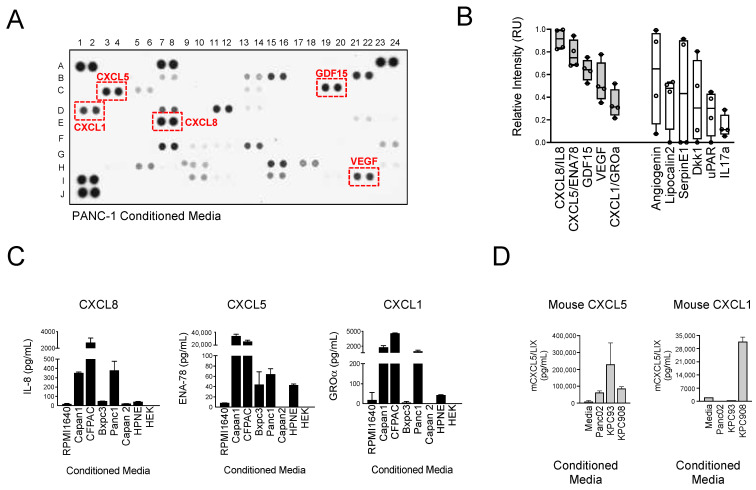
CXCR2 ligands are the most abundant soluble factors secreted by PDAC cells. (**A**) Cytokine arrays were used to identify relative concentrations of 105 secreted factors from four PDAC cell lines (BxPC-3, Capan-1, CFPAC-1, and PANC-1). Concentrations were determined relative to reference spots (A1,2,23,24 and J1,2). A representative cytokine array image using PANC-1 CM is shown. (**B**) Quantification of secreted factors that were upregulated in four out of four (gray) or three out of four (white) cell lines are shown. (**C**) CM from PDAC cell lines, HEK cells, and HPNE cells was analyzed by ELISA for CXCL8, CXCL5, and CXCL1 (*n =* 2–10). (**D**) CM from cell lines established from PDAC mouse models Panc02 and KPC was analyzed by ELISA for mouse CXCL5 and mouse CXCL1 (*n* = 2).

**Figure 2 biomolecules-15-00645-f002:**
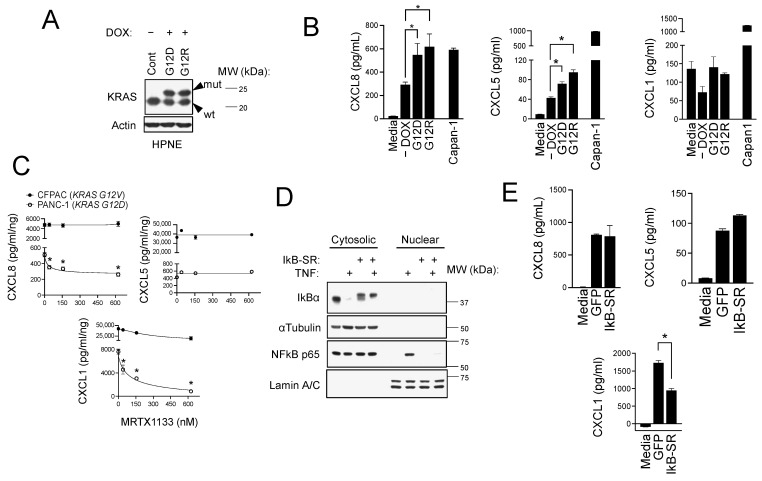
CXCR2 ligand production in PDAC cells is mutant-KRAS- and NFκB-dependent. (**A**) Western blot analysis of KRAS G12D or G12R mutant expression in inducible HPNE cells upon treatment with 1 μg/mL doxycycline for 48 h. (**B**) CM was collected from KRAS-mutant-inducible HPNE cells with or without treatment with doxycycline and analyzed by ELISA for CXCL5 and CXCL1. Statistical significance was determined using ordinary one-way ANOVA, and (-) DOX, G12D, and G12R were compared by Šídák’s multiple comparisons test (* *p* < 0.05, *n* = 3). (**C**) PANC-1 cells and CFPAC-1 cells were treated with a dose course of MRTX1133 for 30 h, and CM was collected and analyzed by ELISA for CXCL8 and CXCL1. Statistical significance was determined using Student’s *t*-test (* *p* < 0.05, *n* = 2). (**D**) PANC-1 cells expressing IκB-SR were treated with 10 ng/mL TNFα for 30 min. Cells were lysed and separated into the cytosolic and nuclear fractions. Western blotting analysis is shown. (**E**) CM was collected from PANC-1 cells expressing GFP or IκB-SR and analyzed by ELISA for CXCL1 and CXCL5. Statistical significance was determined using ordinary one-way ANOVA, and GFP and IkB-SR were compared by Šídák’s multiple comparisons test (* *p* < 0.05, *n* = 3). Original images of (**A**,**D**) can be found in Appendix A.

**Figure 3 biomolecules-15-00645-f003:**
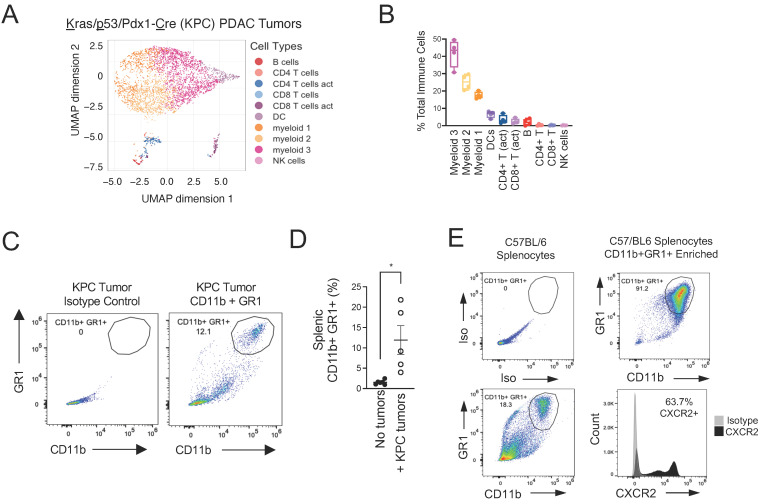
MDSCs in mice with PDAC are CXCR2-positive. (**A**) C57BL/6 mice were injected subcutaneously with KPC cells, and tumors and spleens were harvested. The composition of tumor-infiltrating immune cells was analyzed by CyTOF. (**B**) Quantification of each cell type in (**A**) is shown. (**C**) Single-cell suspensions were prepared from harvested KPC tumors and were analyzed by flow cytometry for tumor infiltrating MDSCs (CD11b+/GR1+). (**D**) Spleens from healthy mice (No tumors) and mice bearing subcutaneous KPC tumors (+KPC tumors) were analyzed by flow cytometry for splenic MDSCs. Statistical significance was determined using Student’s *t*-test (* *p* < 0.05, *n* = 6 [No tumors], *n* = 5 [+KPC tumors]). (**E**) MDSCs were isolated from spleens by negative selection and analyzed by flow cytometry for CD11b, GR1, and CXCR2.

**Figure 4 biomolecules-15-00645-f004:**
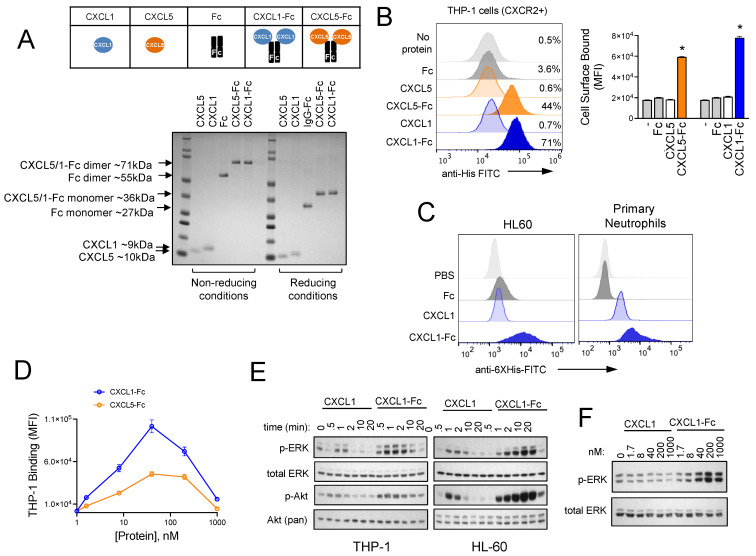
Design and characterization of CXCR2 ligand fusion proteins. (**A**) Purified CXCL1, CXCL5, Fc, and fusion proteins CXCL1-Fc and CXCL5-Fc were analyzed by SDS-PAGE. Proteins were separated under reducing and non-reducing conditions and visualized using Coomassie Blue staining. (**B**) THP-1 cells were incubated on ice with proteins for 30 min, and binding was analyzed by flow cytometry using a FITC-conjugated anti-His tag antibody. Statistical significance was determined using Student’s *t*-test (* *p* < 0.05, *n* = 3). (**C**) HL60 and primary human neutrophils were incubated with proteins as in (**B**) and analyzed by flow cytometry, as in (**B**). (**D**) Binding of CXCL1-Fc and CXCL5-Fc to THP-1 cells was analyzed as in (**B**) across a range of concentrations. (**E**) THP-1 cells and HL60 cells were incubated with CXCL1 or CXCL1-Fc for the indicated times. Cell lysates were analyzed by Western blotting for p-ERK, total ERK, p-AKT, and total AKT (pan). (**F**) THP-1 cells were incubated with the indicated concentrations of CXCL1 or CXCL1-Fc for 5 min. Cell lysates were analyzed by Western blotting for p-ERK and total ERK. Original images of (**E**,**F**) can be found in Appendix A.

**Figure 5 biomolecules-15-00645-f005:**
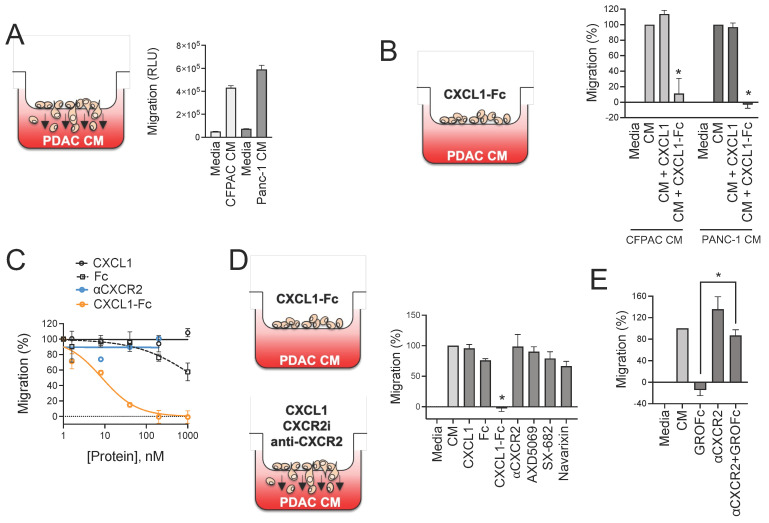
CXCL1-Fc superkine disrupts chemotaxis of myeloid cells. (**A**) Migration assays using a Boyden chamber design were performed to analyze the migration of myeloid cells (THP-1) towards PDAC CM. THP-1 myeloid cells were seeded in the top well of an 8 μm porous membrane insert. The bottom well contained PANC-1 CM, CFPAC-1 CM, or DMEM + 10% FBS as a control. Cells were incubated for 2 h, and migrated cells were quantified using a CellTiter Glo assay. Data are displayed as relative luminescence units (RLUs). (**B**) Either 100 nM CXCL1 or CXCL1-Fc was added to the top well with cells, and migration was analyzed as in (**A**). Data are shown as % migration normalized to untreated controls. Statistical significance was determined using a one-sample *t*-test against a hypothetical mean of 100% (* *p* < 0.05, *n* = 3). (**C**) Cells in the top well were treated with a dose range of CXCL1, Fc, anti-CXCR2 neutralizing antibody, or CXCL1-Fc. Migration towards PANC-1 CM was analyzed as in (**A**). (**D**) Cells in the top well were treated with 5 μM CXCR1/2 inhibitors AZD5069, SX-682, and Navarixin, and migration towards PANC-1 CM was analyzed as in (**C**). Statistical significance was determined using a one-sample *t*-test against a hypothetical mean of 100% (* *p* < 0.05, *n* = 3). (**E**) Cells in the top well were treated with CXCL1-Fc, anti-CXCR2 neutralizing antibody, or both, and migration was analyzed as in (**C**). Statistical significance was determined using ordinary one-way ANOVA, and CXCL1-Fc and CXCL1-Fc + αCXCR2 were compared by Šídák’s multiple comparisons test (* *p* < 0.05, *n* = 4).

**Figure 6 biomolecules-15-00645-f006:**
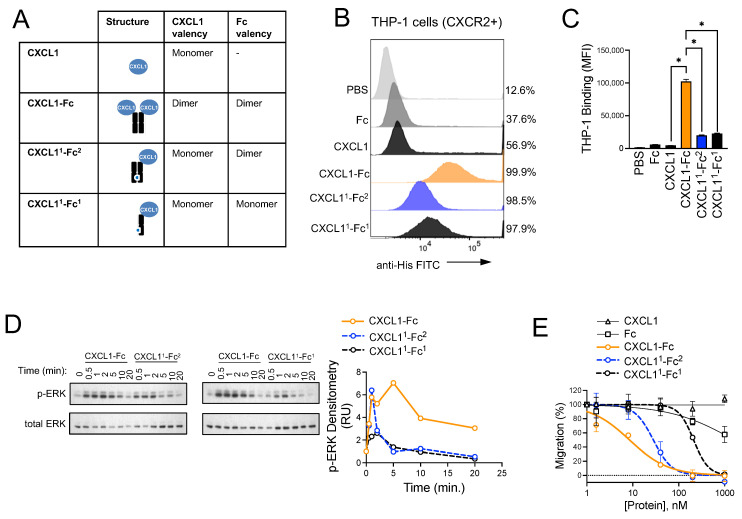
Fc dimerization is critical for the superkine activity of CXCL1-Fc. (**A**) Schematic and characteristics of the CXCL1-Fc variants are shown. Compared to the CXCL1-Fc superkine, which consists of two CXCL1 ligands and an Fc dimer, CXCL1^1^-Fc^2^ consisted of a CXCL1 monomer with an Fc dimer, and CXCL1^1^-Fc^1^ was composed of a CXCL1 monomer and an Fc monomer. (**B**) Binding of Fc, CXCL1, CXCL1-Fc, CXCL1^1^-Fc^2^, and CXCL1^1^-Fc^1^ to THP-1 cells was analyzed by flow cytometry using an FITC-conjugated anti-6X His antibody. (**C**) Mean fluorescence intensity of flow cytometry in (**B**) was quantified. Statistical significance was determined using ordinary one-way ANOVA, and CXCL1, CXCL1-Fc, CXCL1^1^-Fc^2^, and CXCL1^1^-Fc^1^ were compared by Šídák’s multiple comparisons test (* *p* < 0.05, *n* = 3). Statistical tests not shown in (**C**) are CXCL1-Fc vs. CXCL1^1^-Fc^2^ (*p* < 0.05), CXCL1-Fc vs. CXCL1^1^-Fc^1^ (*p* < 0.05), and CXCL1^1^-Fc^2^ vs. CXCL1^1^-Fc^1^ (*p* = 0.1163). (**D**) THP-1 cells were incubated with CXCL1, CXCL1-Fc, CXCL1^1^-Fc^2^, or CXCL1^1^-Fc^1^ for the indicated times, ranging from 0.5–20 min. Downstream cell signaling events were analyzed by Western blotting using p-ERK as the read-out. Total ERK is included as a loading control. p-ERK signal was normalized to total ERK and plotted over time in an x-y plot. (**E**) Migration assays were performed to analyze the inhibition of THP-1 cell migration toward PDAC CM by CXCL1, Fc, CXCL1-Fc, CXCL1^1^-Fc^2^, and CXCL1^1^-Fc^1^ (*n* = 2). Original images of (**D**) can be found in Appendix A.

**Figure 7 biomolecules-15-00645-f007:**
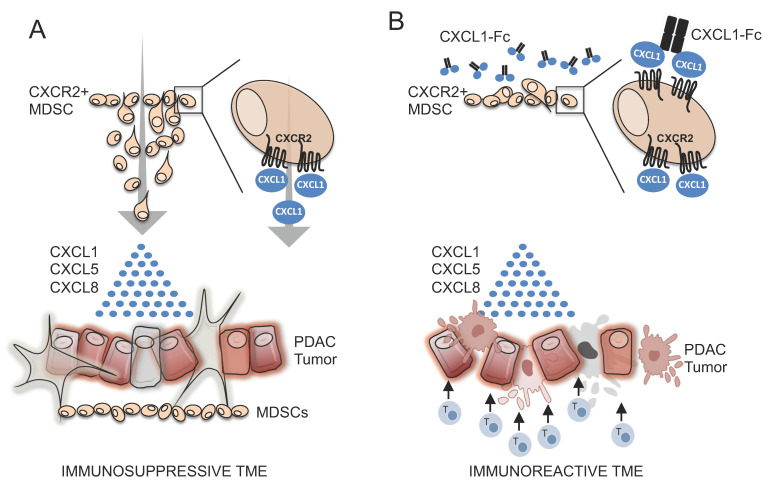
Model of CXCL1-Fc superkine inhibition of myeloid cell infiltration of PDAC tumors. (**A**) MDSCs migrate into PDAC tumors along chemotactic gradients composed primarily of CXCR2 ligands. In the PDAC TME, MDSCs suppress immune effector cells and protect tumors from host immune surveillance. (**B**) CXCL1-Fc superkine induces robust CXCR2 signaling and drowns out chemotactic signals from PDAC tumors and prevents MDSC infiltration.

## Data Availability

The original contributions presented in this study are included in the article. Further inquiries can be directed to the corresponding author.

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
