# Peer review of "Modulating the CXCR2 Signaling Axis Using Engineered Chemokine Fusion Proteins to Disrupt Myeloid Cell Infiltration in Pancreatic Cancer"

_biomolecules, 2025, doi:10.3390/biom15050645_

Round 1

Reviewer 1 Report

Comments and Suggestions for Authors

The manuscript submitted by Khristopher et al. describes the differential release of CXCR2 ligands by PDAC cells, which are partly dependent on mutant KRAS and NFκB signaling. Due to the role of these ligands in attracting MDSCs and their contribution to sustaining an immunosuppressive microenvironment, the authors designed CXCR2 ligand-Fc fusion proteins, or "superkines," and demonstrated their efficacy in inhibiting MDSC migration. The authors suggest that their findings could improve immunotherapy outcomes for PDAC patients by disrupting myeloid cell recruitment to tumors.

Here some points:

-Introduction

In this section, additional information, and relevant literature,  is needed about Cytokine-Fc fusion proteins, including their structural design, molecular mechanisms, production, and their diverse therapeutic applications focusing on oncology.

Materials and methods are clear, detailed and well-explained. In migration assays,  the pore size of ThinCert Cell Culture Inserts should be added, as mentioned in results.

-Results

Figure 1C could benefit from a statistical analysis comparing cytokine levels in the conditioned medium of PDAC cells in respect to HPNE cells, as this would enhance data visualization and improve the clarity of interpretation.

Results displayed in Figure 2C could be improved by using a pan-KRAS inhibitor (e.g.the molecule. BI-2493) in relevant cell lines (i.e. cell lines highly secreting CXCL1, CXCL-8 and CXCL5) at least human cell lines.

Author Response

Please see attachment. All changes are highlighted in yellow in the “manuscript v4” document.

Reviewer 2 Report

Comments and Suggestions for Authors

The authors address an interesting question about the modulation of the CXCR2 signaling axis using engineered chemokine fusion proteins to disrupt myeloid cell infiltration in pancreatic cancer. The topic is relevant but the paper lacks scientific rigor, therefore major revision is needed. 

Results 1:

- The authors say CXCR2 ligands are in the top 5 most expressed cytokines but CXCL1/GROa is not in the top 5 FigS1.

- Fig S1 is written to represent relative expression. But what has it been normalized to?

- Missing N in Fig 1C and D

- Missing ELISA of CXCL8 in PDAC Mouse lines. Perhaps it is not secreted?(Fig 1D)

- Cytokines are much more concentrated in conditioned media from human cells than murine cells. What could this mean?

Results 2:

- Fig 2B,C,E: The authors do the statistical test with N=2!!! and say the differences are significant!

- Fig 2B: Why isn't CXCL8 also looked at? In mutated HPDE the authors look at CXCL5 and CXCL1 while in PANC and CFPAC treated with MRTX1133 they look at CXCL1 and CXCL8. I don't understand why they don't always look at all 3 cytokines.

In HPNE cells in which Kras is mutated, there is an  increase CXCL5 but not CXCL1. But when the authors take human lines mutated for Kras and treat them with the inhibitor MRTX1133 they only show the effect on CXCL1 and CXCL8. I would have expected a greater effect on CXCL5 but it is not shown.

- What is the effect of IKB-SR on CXCL8?

Results 3:

- Fig 3D: it says N=6 but N=5

- Fig3E - it says there is high expression of CXCR2 in MDSCs in tumours but compared to what? The authors only compare marked from unmarked so they can only conclude that it is expressed.

- Fig 3C: as I understand,  they authors isolate the MDSCs from the tumour using a kit and then mark them with CD11b and Gr1 to verify purification (see materials and methods). But the cells positive for CD11b and Gr1 are only 12%. Maybe there is something wrong?

- It would have been interesting to look at CXCR2 in MDSCs in the tumour and not just in the spleen

Results 4:

- Fig4B: Statistics with N=2 ??

- Fig4B: From the materials and methods it seems that Fc blocking is not done. Could it be that the result is due to binding of the Fc portion to Fc receptors? THP-1 are monocytes... do they express the Fc receptor?

- Fig 4C-D-E is missing quantification.

- Fig S4B- no quantification

Results 5:

- Fig 5A,B,C,D: do the statistic with N=2!!!

- Fig 5D - missing significance in the graph

- Fig S5 - missing quantification statistic. Looks like the result is due to one blot. Normalizer is missing. P-Erk has two bands (the two phosphorylations), has it been normalised to the total? Would be to see both phosphorylations if they change.

Results 6:

- Fig 6C - what is the N? Why is there no statistic?

- Fig 6D - the quantification is missing

- Fig6E - missing N and quantification

- It would be necessary to show that Fc-CXCL1 is not phagocytosed by macrophages that recognise the Fc portion? One could show with immunofluorescence using anti-CXCL1 that Fc-CXCL1 is not internalised in THP-1 cells.

- It would be interesting to inject Fc-CXCL1 into mice with tumours to see if it is accumulated in the tumour or elsewhere. It would also be interesting to see if there is an effect in tumour growth.

Author Response

(The authors gave the same response as above.)

Reviewer 3 Report

Comments and Suggestions for Authors

This manuscript investigates the CXCR2 signaling axis in the migration of myeloid cells in pancreatic ductal adenocarcinoma (PDAC). The data reveal that CXCR2 ligand expression by PDAC is dependent on mutant KRAS and NF-κB signaling, which leads to the recruitment of myeloid cells. The use of a CXCR2 ligand-Fc fusion protein effectively suppresses the migration of myeloid cells in a PDAC-conditioned medium. However, antibodies against CXCR2 and inhibitors targeting CXCR2 signaling fail to suppress myeloid cell migration. Detailed comments and suggestions are listed below:

  1. The figure legend for Figure 1A and B does not match the images.
  2. Figure 2: The results show that CXCL5 and CXCL1 expression is upregulated in cells transduced with mutant KRAS. What about the expression of CXCL8? Additionally, how does CXCL5 expression change after treatment with MRTX1133?
  3. Please explain why the cells were treated with Dox in Figure 2A and B.
  4. Figure 3: What are the differences between myeloid 1, 2, and 3 cells?
  5. The results in Figure 4B demonstrate that both CXCL5-Fc and CXCL1-Fc bind strongly to THP-1 cells. Why was CXCL1-Fc chosen for the experiment in Figure 4C–E instead of CXCL5-Fc?
  6. Figure 4E is not described in the figure legend.

Author Response

(The authors gave the same response as above.)

Reviewer 4 Report

Comments and Suggestions for Authors

The article takes a very important and interesting approach to the treatment of one of the most difficult to cure cancers, which is PDAC.

In such an approach to the study, it is important to have the most comprehensive view of this disease entity.

In the section on flow cytometry, I would recommend separating the antibodies that were sought from the isotype antibodies.

Why were only splenocytes examined in the study and the researchers were not interested in examining TLI’s since they are the most important in creating TME?

The materials and methods mention the use of one cell line, while the figures list 7 cell lines (Figure 1).

The images from the western blot (Figure 2) lack molecular weight measurement.

Apart from that, it is a very well-written article, with well-chosen research methods.

Author Response

(The authors gave the same response as above.)

Round 2

Reviewer 1 Report

Comments and Suggestions for Authors

Authors have properly addressed the referees’ comments.

Reviewer 3 Report

Comments and Suggestions for Authors

The authors have addressed all the questions raised by this reviewer.